# Cross-sectional survey of child weight management service provision by acute NHS trusts across England in 2020/2021

Ruth Mears ![ORCID] ,[1,2] Sofia Leadbetter,[3,4] Toby Candler,[3,4] Hannah Sutton,[4] Deborah Sharp ![ORCID] ,[1] Julian P H Shield ![ORCID] [3]

¹Centre for Academic Primary Care, Bristol Medical School, University of Bristol, Bristol, UK
²Centre for Exercise Nutrition and Health Sciences, University of Bristol, Bristol, UK
³National Institute for Health Research Bristol Biomedical Research Centre, University Hospitals Bristol and Weston NHS Foundation Trust, Bristol, UK
⁴Faculty of Health Sciences, University of Bristol, Bristol, UK

**Correspondence to**
Dr Ruth Mears;
rm14101@bristol.ac.uk

## ABSTRACT

**Objective** With one in five children in England living with obesity, we mapped the geographical distribution and format of child weight management services provided by acute National Health Service (NHS) trusts across England, to identify breadth of service provision.

**Design** A cross-sectional survey.

**Setting** The survey was sent to acute NHS trusts (n=148) in England in 2020, via a freedom of information request.

**Participants** Responses were received from 139 of 148 (94%) acute NHS trusts, between March 2020 to March 2021.

**Outcome measures** The survey asked each acute NHS trust whether they provide a weight management service for children living with obesity. For those trusts providing a service, data were collected on eligibility criteria, funding source, personnel involved, number of new patients seen per year, intervention duration, follow-up length and outcome measures. Service characteristics were reported using descriptive statistics. Service provision was analysed in the context of ethnicity and Index of Multiple Deprivation score of the trust catchment area.

**Results** From the 139 survey respondents, 23% stated that they provided a weight management service for children living with obesity. There were inequalities in the proportion of acute NHS trusts providing a service across the different regions of England, ranging from 4% (Midlands) to 36% (London). For trusts providing a service, there was variability in the number of new cases seen per year, eligibility criteria, funding source, intervention format and outcome measures collected. A multidisciplinary approach was not routinely provided, with only 41% of services reporting ≥3 different staff disciplines.

**Conclusion** In 2020/2021, there were geographical inequalities in weight management service provision by acute NHS trusts for children living with obesity. Services provided lacked standardisation, did not routinely offer children multidisciplinary care and were insufficient in size to meet need.

## STRENGTHS AND LIMITATIONS OF THIS STUDY

⇒ There are no previous studies in the research literature exploring the geographical equity and format of child weight management services provided by acute National Health Service (NHS) trusts across England.

⇒ The survey response rate (94%) was good, allowing for an accurate oversight of service provision by acute NHS trusts in England.

⇒ Outcome data for services were not collected, meaning service effectiveness could not be assessed.

⇒ Ethnicity and measures of deprivation data for the individuals accessing the services within each acute NHS trust were not collected, thereby limiting ethnicity and deprivation comparisons to acute NHS trust catchment area-level data.

hypertension, metabolic syndrome, nonalcoholic fatty liver disease and obstructive sleep apnoea.[1 2] Around 55% of children with obesity will remain obese as adolescents, and 80% of adolescents with obesity will continue to be obese as adults.[3] Obesity has both direct healthcare-related costs and indirect costs to society through short-term and long-term inability to work.[4 5] In 2018, obesity cost the UK an estimated £60 billion (3% of its GDP).[6 7]

In the UK, child weight management services are defined as: tier 1 (obesity prevention), tier 2 (lifestyle weight management services), tier 3 (specialist obesity services) and tier 4 (pharmacological or surgical obesity services).[8] Tier 1 and tier 2 services are commissioned by Public Health working within Local Authorities. In theory, tier 3 and tier 4 services cater for children with severe or complex obesity, or special needs, and are commissioned by Clinical Commissioning Groups (CCGs) working within the National Health Service (NHS). There is no mandatory requirement for local authorities or the NHS to commission child weight management services.

## INTRODUCTION

Obesity in childhood and adolescence represents a major public health challenge with short and long-term consequences at both an individual and societal level. Children with obesity are more likely to have type 2 diabetes, dyslipidaemia,

Commissioning guidance for specialist services for children with severe or complex obesity, recommend a multidisciplinary team (MDT) approach, including a paediatrician, dietitian, nurse, physiotherapist, psychologist, liaison psychiatrist and MDT access to a social worker.[9] Previous studies have explored the nature and provision of tier 2 weight management services across England, but there is no documentation within the research literature to gain national oversight of specialist obesity weight management services provided by acute NHS trusts in England.[10 11] A Public Health England (PHE) mapping study of tier 3 weight management services for children in 2015 reported a poor response rate limiting accurate representation of service provision at the time.[11] The lack of understanding about current provision of specialist obesity services was highlighted in the Chief Medical Officer's report on childhood obesity.[7]

Our local centre has been approached by many teams across England and Wales seeking consultations with our service as local services were not commissioned, suggesting that there is no equity of access across the country. The primary objective of the study was to determine how many acute NHS trusts provide a weight management service for children, and the geographical distribution of these services. The secondary objective of the study was to understand the nature of weight management services provided by acute NHS trusts, in terms of eligibility criteria, funding source, whether an MDT approach operated, the reach of the service and details regarding the intervention and outcome data collected. The final objective was to understand whether there were any inequalities in ethnicity and deprivation scores between the catchment populations of NHS trusts providing a weight management service, compared with those that did not.

## METHODS
### Design of the survey
The survey comprised eight questions and was developed by JPHS based on his experience working within a paediatric obesity specialist clinic at the Bristol Royal Hospital for Children, University Hospitals Bristol & Weston NHS Foundation Trust (online supplemental file 1). All acute NHS trusts surveyed in England were asked, 'Does your trust provide a weight management service for children living with obesity?'. Those affirming they did, were asked questions regarding the service's eligibility criteria, source of funding, key personnel involved, number of new patients seen per year, duration of their intervention, length of follow-up and outcome data collected by their service. Questions were either free-text or multiple choice (single or multiple answer questions).

### Recruitment and characteristics of acute NHS trust participants
The survey was sent by email to each acute NHS trust Freedom Of Information (FOI) office across England

(n=148) as an FOI request with a fillable word document for ease of use, to be emailed back to the research team. Where a response was not received by the acute NHS trust within 20 working days, a further reminder email was sent. If no response was received, JPHS contacted potential service leads from a list provided by the Children and Young People Transformation Programme Team at NHS England.

The 'PHE NHS acute trust catchment population tool' was used to obtain data (where available) about the catchment population for each acute NHS trust, in terms of Index of Multiple Deprivation (IMD) score and ethnicity.[12] This tool also provided details regarding the acute NHS trust type (small, medium, large, multiservice, specialist and teaching).

### Data analysis
Data were collated and analysed in Excel V.2206 and Stata V.17. In order to understand potential inequalities in the geographical distribution of weight management services for children, trusts were organised into the seven regions of England (North East and Yorkshire, North West, South East, South West, London, East of England and Midlands) and the proportion of trusts within each geographical region providing a service was quantified. The mean IMD score and mean proportion of patients within each different ethnicity category were compared between those trusts providing and not providing a service. Comparisons between the proportion of trusts offering and not offering a service were also made according to acute NHS trust type. For acute NHS trusts providing a weight management service, the characteristics of the service provided were reported using descriptive statistics.

### Patient and public involvement
Patients or the public were not involved in the design, or conduct, or reporting, or dissemination plans of our research.

## RESULTS
Responses were received between March 2020 and March 2021, from 139 of 148 acute NHS trusts contacted (94%). From the 139 respondents, 32 trusts (23%) stated that they provide a weight management service for children living with obesity. Some trusts offered more than one service for children living with obesity. The mean IMD score and ethnicity of patients were similar between those within a catchment area of an acute NHS trust providing and not providing a weight management service (online supplemental file 2).

### Location in England and acute NHS trust type
The greatest proportion of trusts providing a weight management service were within London (36% of trusts). In contrast, only 4% of trusts in the Midlands provided a service (table 1). Multiservice and teaching trusts were more likely to provide a weight management service than other acute NHS trusts (table 2).

**Table 1** Geographical distribution of services across England

| | Service provided (n=no of acute NHS trusts) | No service provided (n=no of acute NHS trusts) | No response to survey (n=no of acute NHS trusts) |
|---|---|---|---|
| North East and Yorkshire | 7 (31.8%) | 15 (68.2%) | 0 (0%) |
| North West | 4 (17.4%) | 17 (73.9%) | 2 (8.7%) |
| South East | 4 (21.1%) | 15 (78.9%) | 0 (0%) |
| London | 9 (36.0%) | 14 (56%) | 2 (8%) |
| East of England | 3 (16.7%) | 14 (77.8%) | 1 (5.6%) |
| Midlands | 1 (4.2%) | 20 (83.3%) | 3 (12.5%) |
| South West | 4 (23.5%) | 12 (70.6%) | 1 (5.9%) |
| *Total* | 32 (22%) | 107 (72%) | 9 (6%) |

NHS, National Health Service.

## Site of intervention, eligibility criteria, funding source and staffing

The characteristics of weight management services, provided by acute NHS trusts across England, are summarised in table 3. Over half of weight management services were hospital-based only (59%). There was considerable variation in the source of funding for these services, with 25% of trusts receiving funding from multiple sources. CCGs were the most frequently cited source of funding.

Eligibility criteria for services were mainly based around a measure of weight (eg, body mass index (BMI), BMI percentile, BMI z-score) but some services reported less specific eligibility criteria (eg, 'all referrals where a dietetic or nutrition related problem has been identified by a medic'). Dietitians were the most frequently reported personnel involved in service provision, followed by paediatricians. Most services did not provide an MDT approach, with 31% of services reporting only one type of staffing personnel involved.

## Characteristics of services provided by acute NHS trusts

Table 4 summarises details of the interventions provided by trusts. There was variation in the number of new cases seen per year, with 59% of acute NHS trusts reporting that their service(s) saw ≤100 new cases per year. Many trusts reported that the intervention and follow-up period were

tailored to the individuals needs rather than following a prespecified pattern. Outcome measures most frequently collected were BMI, BMI z-score/SDS or BMI centile. A variety of other outcome measures were reported, including a parent or user satisfaction survey, changes to emotional eating behaviours, reduction of inequalities among targeted groups and 'Systemic Clinical Outcome and Routine Evaluation' (SCORE-15) index of family functioning and change.[13]

## DISCUSSION
## Main findings

There were significant gaps in the provision of child weight management services by acute NHS trusts across England, with 77% of survey respondents not providing a service. Within England, there was considerable variation in the distribution of services with a greater proportion of acute NHS trusts providing a service in certain geographical regions (eg, London 36% and the North-East and Yorkshire 32%) compared with others (eg, Midlands 4%). There was a lack of consistency in the funding source and eligibility criteria for children to access services between different acute NHS trusts. A multidisciplinary approach was not routinely provided and there was no standardisation of the service provided between different trusts

**Table 2** Service provision according to acute NHS trust type

| | Service provided (n=no of acute NHS trusts) | No service provided (n=no of acute NHS trusts) | No response to survey (n=no of acute NHS trusts) |
|---|---|---|---|
| Acute—small | 6 (19.4%) | 24 (77.4%) | 1 (3.2%) |
| Acute—medium | 6 (19.4%) | 23 (74.2%) | 2 (6.5%) |
| Acute—large | 5 (18.5%) | 20 (74.1%) | 2 (7.4%) |
| Acute—multiservice | 2 (66.7%) | 1 (33.3%) | 0 (0%) |
| Acute—specialist | 3 (18.8%) | 13 (81.3%) | 0 (0%) |
| Acute—teaching | 10 (29.4%) | 22 (64.7%) | 2 (5.9%) |
| Total | 32 (23%) | 103 (73%) | 7 (5%) |

Data regarding the type of acute NHS trust was available for 142 trusts.
NHS, National Health Service.

**Table 3** Characteristics of weight management services provided by acute NHS trusts

| | No of acute NHS trusts (n=32) |
|---|---|
| Location of services provided by each acute NHS trust | |
| Hospital | 19 |
| Community | 8 |
| Hospital and community based | 4 |
| Not specified | 1 |
| Source of funding* | |
| Clinical Commissioning Group | 15 |
| Trust self-funding | 9 |
| NHS England | 2 |
| County Council, Local Authority or Public Health | 7 |
| Other | 7 |
| Not specified | 1 |
| Eligibility criteria† | |
| BMI >91st centile | 5 |
| BMI >91st centile and another criterion | 4 |
| BMI >98th centile | 4 |
| BMI >98th centile and another criterion | 9 |
| BMI >99.6th centile | 4 |
| BMI >99.6th centile and another criterion | 3 |
| BMI SD Score (SDS) >2.2 | 1 |
| BMI SDS ≥3.33 in child over 2 years old | 1 |
| BMI SDS >3.5 or BMI SDS >3.0 with comorbidities | 2 |
| Other | 18 |
| Staff disciplines involved in service provision for each acute NHS trust‡ | |
| Dietitians | 27 |
| Paediatricians | 16 |
| Hospital or community-based nursing staff | 12 |
| Psychologists | 7 |
| Exercise specialists | 5 |
| Children and Adolescent Mental Health Services | 3 |
| Social workers | 2 |
| Not specified | 2 |
| Trained volunteers | 1 |
| No of staff disciplines providing service for each acute NHS trust | |
| 1 | 10 |
| 2 | 7 |
| 3 | 6 |
| 4 | 5 |

Continued

**Table 3** Continued

| | No of acute NHS trusts (n=32) |
|---|---|
| 5 | 1 |
| 6 | 1 |
| Not specified | 2 |

*Some acute NHS trusts received funding from more than one source for their service(s).
†Some acute NHS trusts reported more than one different eligibility criteria for their service(s).
‡Some acute NHS trusts reported more than one different staff disciplines involved in their service provision.
BMI, body mass index; NHS, National Health Service.

in terms of length of intervention, follow-up period and outcome measures.

### Strengths and weaknesses

This is the first study in the research literature to explore the nature of child weight management service provision by acute NHS trusts across England. The response rate by acute NHS trusts was good (94%). As acute NHS trusts are typically responsible for the provision of tier 3 weight management services, this paper provides a good indicator of what services are available for children with severe obesity. It was not within the remit of this study to capture non-NHS tier 2 weight management programmes. It was also not possible to merge the data on NHS provision with pre-existing datasets about non-NHS tier 2 weight management provision (ie, those commissioned by local authorities), partly due to the rapid commissioning and decommissioning of the latter.[10]

Data detailing intervention components and outcome measures were not requested to reduce burden on individual trust's FOI response teams and improve overall response rate.[14] Thus, it was not possible to assess the relative effectiveness of different interventions offered by each acute NHS trust. However, given the lack of standardisation in eligibility criteria and outcome measures across different NHS trusts, meaningful comparative analyses are unlikely to have been possible. Deprivation and ethnicity data for those actually accessing each service were not collected, so comparisons in these measures were limited to acute NHS trust population catchment area data. The newly commissioned 'Complications from Excess Weight' (CEW) clinics are collecting robust process and outcome data as a national audit and we envisage that all child weight management services will be encouraged to report similar data in the future.[15]

Data were collected through a FOI request meaning it is not possible to ascertain who supplied survey data for each NHS trust. The assumption is that the FOI request would have been appropriately handed to the lead for the clinical service.

| Table 4 | Details of the services provided by acute NHS trusts |
|---|---|

| | No of acute NHS trusts (n=32) |
|---|---|
| **Total no of new cases seen per year for each acute NHS trust** | |
| ≤50 | 11 |
| 51–100 | 8 |
| 101–150 | 4 |
| 151–200 | 2 |
| 201–250 | 0 |
| 251–300 | 0 |
| 301–350 | 2 |
| Not specified | 5 |
| **Length of intervention*** | |
| 0–6 months | 9 |
| 6–12 months | 4 |
| 12–18 months | 1 |
| Tailored to the individual | 11 |
| Not clearly specified | 10 |
| Other | 2 |
| **Usual follow-up period*** | |
| 0–6 months | 6 |
| 6–12 months | 5 |
| 12–18 months | 6 |
| Over 18 months | 4 |
| Tailored to the individual | 9 |
| Not clearly specified | 3 |
| Other | 1 |
| **Outcome measures** | |
| BMI | 20 |
| BMI-SDS or BMI z-score | 12 |
| BMI-centile | 2 |
| Weight | 7 |
| Waist circumference | 2 |
| Body composition | 3 |
| Blood pressure | 2 |
| Sleep study results | 3 |
| Blood tests | 4 |
| Imaging | 1 |
| Changes to diet | 7 |
| Changes to physical activity levels | 4 |
| Changes to mental well-being | 7 |
| Quality of life | 2 |
| Other | 12 |

Respondents sometimes detailed more than one clinical service in the trust so n responses may be greater than 32.
NHS, National Health Service.

## Contextualisation and implications of findings

A lack of adequate multidisciplinary weight management clinics for children with obesity and unequal geographical distribution of available services has also been described in Australia.[16 17] This contrasts with current guidelines and expert recommendations for obesity management in children, which advises an MDT approach to ensure optimal care.[9 18 19] Similar to our study, data reported from Canadian paediatric weight management services has shown variability in eligibility criteria, clinic size and intervention length.[20]

In England (2020/2021), 14% of children aged 4–5 years and 26% of children aged 10–11 years were identified as obese by the National Child Measurement Programme (NCMP).[21] Therefore, the current 32 sparsely distributed acute NHS trust services in England (generally seeing <200 new cases per year) will only reach a minute fraction of children living with obesity (or severe obesity).

A pathway modelling study of National Institute for Health and Care Excellence guidance against Health Survey for England data, suggested that there is a 'need for consistent evidence-based commissioning of services across the childhood obesity pathway based on population burden', however, our study indicates this is still not happening.[22] Despite the West Midlands having the highest prevalence of severe obesity according to 2019/2020 reception NCMP data (3.1% vs England average of 2.5%), the Midlands region had the lowest proportion of NHS trusts providing a weight management service (n=1/24, 4.2%).[21]

The NHS Long Term Plan aims to treat a further 1000 children a year for severe complications related to their obesity through 'CEW' clinics. These CEW clinics have a universal data collection system for patient demographics and outcomes to allow comparison across sites and understand the reach of the service according to ethnicity and deprivation. However, with previous estimates from 2014 suggesting that 556 000 children would be eligible for assessment in secondary care, this planned expansion of services will still not reach the majority of children who may require treatment.[22 23]

The lack of consistent sources of funding for specialist obesity clinics in acute NHS trusts likely contributes to the inequalities in the geographical distribution of services. Clearly defined routes of funding for child weight management clinics in acute NHS trusts are needed to help minimise inequalities in service provision. Furthermore, given the 13 percentage-point gap in obesity rates between the most deprived and least deprived children in 2019, services need to ensure that within the geographical areas they provide for, they are reaching the most deprived children.[24]

## CONCLUSION

There needs to be a clear, realistic national strategy outlining who should receive initial priority for specialist multidisciplinary obesity care, as the need for support

will likely exceed capacity for the foreseeable future and service provision across England should be equitable. The CEW clinics may help to reduce current inequalities in service provision and target those at greatest risk from obesity. However, significant gaps in support for children who are obese but do not meet the CEW clinic criteria will likely remain, especially given the variable tier 2 service provision across England.[10] Furthermore, as referral into the CEW clinic will be reliant on the presence of comorbidities, consideration needs to be given to determine the most appropriate way of ensuring that children living with obesity (or at least severe obesity) are assessed for comorbidities and thereby given the opportunity to access specialist care where needed.

**Contributors** JPHS designed the study. TC, SL and HS collected the data. RM and JPHS analysed the data and drafted the initial manuscript. JPHS, TC, SL, HS and DS revised the final manuscript. JPHS is guarantor for this study.

**Funding** RM is funded by a National Institute for Health Research (NIHR) In-Practice Fellowship Award (NIHR-IPF-16-10-07) for this research. The NIHR Biomedical Research Centre at University Hospitals Bristol and Weston NHS Foundation Trust supports Professor Julian Hamilton-Shield.

**Disclaimer** This publication presents independent research funded by the National Institute for Health Research (NIHR). The views expressed in this publication are those of the authors and not necessarily those of the NIHR.

**Competing interests** JPHS is clinical lead for a tier 3 childhood obesity service in Bristol and will be a hub for the new 'Complications from Excess Weight' (CEW) clinics funded by NHS England.

**Patient and public involvement** Patients and/or the public were not involved in the design, or conduct, or reporting, or dissemination plans of this research.

**Patient consent for publication** Not applicable.

**Provenance and peer review** Not commissioned; externally peer reviewed.

**Data availability statement** All data relevant to the study are included in the article or uploaded as online supplemental information.

**ORCID iDs**
Ruth Mears http://orcid.org/0000-0002-1498-6996
Deborah Sharp http://orcid.org/0000-0003-3071-9860
Julian P H Shield http://orcid.org/0000-0003-2601-7575

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
