## [Reviewer comments · BMJ Open]

ARTICLE DETAILS

TITLE (PROVISIONAL)	A cross-sectional survey of child weight management service provision by acute NHS trusts across England in 2020/21
AUTHORS	Mears, Ruth; Leadbetter, S; Candler, Toby; Sutton, Hannah; Sharp, Deborah; Shield, Julian

VERSION 1 – REVIEW

REVIEWER	Saha, Unatti Tilburg University
REVIEW RETURNED	30-Mar-2022

GENERAL COMMENTS	The manuscript was well written and statistical analyses are mainly descriptive. Author (s) may put the sample size at end of Table title. Future study may aim to measure the outcome of obesity to assess the impact of weight management service provision.
--

REVIEWER	O'Malley, Grace RCSI University of Medicine and Health Sciences, Physiotherapy Department
REVIEW RETURNED	15-Jun-2022

GENERAL COMMENTS	well done on developing and completing this important piece of work which will have direct impact on how services for children with obesity are developed and delivered throughout England. There are a number of items that should be considered in order to strengthen the paper further. Please see below: -Please use child-first language throughout. There is inconsistent use.-On line 40 for readers in the UK and Europe, is the term physical therapist used within the NHS or is physiotherapist the common title? You could check the Health and Care Professions Council for clarification.-in the introduction you highlight the gap in current understanding regarding service provision and then address the objective. Perhaps you could address the reasoning behind why this objective is important. This would help the reader develop an understanding regarding considerations related to equity, equality and whether an integrated model of care for obesity is required or needed.-During recruitment of trusts and the FOI requests, were these submitted using a standard FOI request form, a particular process or
---

	purely a letter to the trust FOI request office? This information would be helpful for the reader in order to better understand how such requests are made across the NHS. -Data analysis: what versions of excel and stata were used? -Did you define 'children'? Perhaps you could include a definition in terms of age and access paediatric healthcare in England as internationally this may differ. -Questionnaire: It is not entirely clear from the supplemental information but was the term obesity defined in the study information leaflet or FOI request? Is there a definition that is commonly used across the acute services in the NHS? If so, please include. -Table 3: the term exercise specialists were used in the survey and paper. Do you know if these are acute service-based gym instructors, PE teachers, exercise physiologists or physiotherapists? For those based outside England and understanding of what professions are regulated by the HCPC or relevant bodies would be useful when comparing service design to international settings. If this detail is unknown perhaps address this in your discussion. -it is important to include a reporting tool for the survey and the CROSS checklist below might be useful to you. if you do not choose to use, please add to your discussion around the development of the survey/FOI questions used and where limitations might exist. https://www.equator-network.org/reporting-guidelines/a-consensus-based-checklist-for-reporting-of-survey-studies-cross/ Thank you for developing this important work. Healthcare services for children with obesity can only improve once it is clear what is currently offered nationally and internationally. This work should assist development of CEW services and help to determine the gaps between Tier 2 and Tier 3 provision.
--	--

VERSION 1 – AUTHOR RESPONSE

Reviewer 1

Dr. Unatti Saha, Tilburg University
 Comments to the Author:

Comment 1: The manuscript was well written and statistical analyses are mainly descriptive. Author (s) may put the sample size at end of Table title.

Response 1: Thank you for taking the time to review our article. We have added totals to Table 1 and Table 2 (final row).

In Tables 3 and 4, the sample size (in terms of number of acute trusts), has been specified.

Comment 2: Future study may aim to measure the outcome of obesity to assess the impact of weight management service provision.

Response 2: Yes, we agree that further comparative analyses are needed to assess the effectiveness of weight management service provision on improving weight status. This is particularly important given the variation in the nature of the format of current service provision which we have identified in our study, so that future commissioning decisions are well-informed.

We have added a further comment in the discussion section as follows:

'Data detailing intervention components and outcome measures were not requested to reduce burden on individual Trust's FOI response teams and improve overall response rate 14. Thus, it was not possible to assess the relative effectiveness of different interventions offered by each acute NHS trust. However, given the lack of standardization in eligibility criteria and outcome measures across different NHS trusts, meaningful comparative analyses are unlikely to have been possible. Deprivation and ethnicity data for those actually accessing each service were not collected so comparisons in these measures were limited to acute NHS trust population catchment area data. The newly commissioned 'Complications related to Excess Weight' (CEW) clinics are collecting robust process and outcome data as a national audit and we envisage that all child weight management services will be encouraged to report similar data in the future'.

Reviewer 2

Ms. Grace O'Malley, Temple Street Children's University Hospital, University College Cork
Comments to the Author:

Comment 1: Dear authors, Well done on developing and completing this important piece of work which will have direct impact on how services for children with obesity are developed and delivered throughout England.

Response 1: Thank you for taking the time to review our article. We hope that it will be helpful to inform future planning of weight management service provision for children.

Comment 2: There are a number of items that should be considered in order to strengthen the paper further. Please see below:

Please use child-first language throughout. There is inconsistent use.

Response 2: We have reviewed throughout the manuscript and changed the following sections to use child-first language:

'Therefore, the current 32 sparsely distributed acute NHS trust services in England (generally seeing <200 new cases per year), will only reach a minute fraction of children living with obesity (or severe obesity)' and:

'Furthermore, as referral into the CEW clinic will be reliant on the presence of co-morbidities, consideration needs to be given to determine the most appropriate way of ensuring that children living with obesity (or at least severe obesity) are assessed for co-morbidities and thereby given the opportunity to access specialist care where needed.'

Comment 3: On line 40 for readers in the UK and Europe, is the term physical therapist used within the NHS or is physiotherapist the common title? You could check the Health and Care Professions Council for clarification.

Response 3: Thank you for highlighting this use of terminology. As physical therapists are more frequently referred to as physiotherapists, we have changed the terminology in our manuscript as follows:

Commissioning guidance for specialist services for children with severe or complex obesity, recommend a multi-disciplinary team (MDT) approach, including a paediatrician, dietitian, nurse, physiotherapist, psychologist, liaison psychiatrist and MDT access to a social worker.

Comment 4: In the introduction you highlight the gap in current understanding regarding service provision and then address the objective. Perhaps you could address the reasoning behind why this objective is important. This would help the reader develop an understanding regarding considerations

related to equity, equality and whether an integrated model of care for obesity is required or needed.
Response 4: We have added a section to address the reasoning behind why this objective is important as follows:

'Our local centre has been approached by many teams across England and Wales seeking consultations with our service as local services were not commissioned, suggesting that there is no equity of access across the country'.

Comment 5: During recruitment of trusts and the FOI requests, were these submitted using a standard FOI request form, a particular process or purely a letter to the trust FOI request office? This information would be helpful for the reader in order to better understand how such requests are made across the NHS.

Response 5: The FOI requests were submitted by email to the trust FOI request office. We have amended the wording as below:

'The survey was sent by email to every acute NHS trust Freedom of Information (FOI) office across England (n=148), as a FOI request with a fillable word document for ease of use, to be emailed back to the research team'.

Comment 6: Data analysis: what versions of excel and stata were used?

Response 6: We have updated this section to include the versions of Excel / Stata used as follows: 'Data were collated and analysed in Excel version 2206 and Stata version 17'.

Comment 7: Did you define 'children'? Perhaps you could include a definition in terms of age and access paediatric healthcare in England as internationally this may differ.

Response 7: In the survey, children were not defined. We asked each acute NHS trust 'Does your trust provide a weight management service for children living with obesity?'

Comment 8: Questionnaire: It is not entirely clear from the supplemental information but was the term obesity defined in the study information leaflet or FOI request? Is there a definition that is commonly used across the acute services in the NHS? If so, please include.

Response 8: The term obesity was not defined in the FOI request. This is because different acute NHS trusts may use differing definitions of obesity and we wanted to capture all services for children living with obesity, provided by acute NHS trusts, and designated by the specific NHS trust as an obesity service.

Comment 9: Table 3: the term exercise specialists were used in the survey and paper. Do you know if these are acute service-based gym instructors, PE teachers, exercise physiologists or physiotherapists? For those based outside England and understanding of what professions are regulated by the HCPC or relevant bodies would be useful when comparing service design to international settings. If this detail is unknown perhaps address this in your discussion.

Response 9: Exercise specialists were not defined. They may have included any of the above and no further details were sought.

Comment 10: it is important to include a reporting tool for the survey and the CROSS checklist below might be useful to you. if you do not choose to use, please add to your discussion around the development of the survey/FOI questions used and where limitations might exist.

<https://www.equator-network.org/reporting-guidelines/a-consensus-based-checklist-for-reporting-of-survey-studies-cross/>

Response 10: Thank you for highlighting this. We have included the RECORD checklist as per the Editorial team's request.

Comment 11: Thank you for developing this important work. Healthcare services for children with obesity can only improve once it is clear what is currently offered nationally and internationally. This

work should assist development of CEW services and help to determine the gaps between Tier 2 and Tier 3 provision.

Response 11: We really appreciate you taking the time to review our paper and we hope that this does assist in informing future service development.

VERSION 2 – REVIEW

REVIEWER	O'Malley, Grace RCSI University of Medicine and Health Sciences, Physiotherapy Department
REVIEW RETURNED	05-Sep-2022
GENERAL COMMENTS	Thank you for addressing the points raised in my review. The paper will be very helpful to clinicians, policy makers and parents interested in the field of obesity treatment. With regards to staffing involved in community-based and hospital-based services it would be prudent when establishing the new CEWs clinics to ensure titles of respective MDT members and relevant accreditation is in place via the HPC (e.g. exercise specialist v exercise physiologist v physiotherapist v physical therapist). From a HR perspective and economic modelling etc it will be important to use the relevant title to ensure accurate unit costs for staff time. Good luck with the next phase of this work! Thanks.